# Use of Topographic Models for Mapping Soil Properties and Processes

Xia Li [1,*] , Gregory W. McCarty [1,*], Ling Du [1] and Sangchul Lee [1,2]

1   Hydrology and Remote Sensing Laboratory, USDA-ARS, Beltsville, MD 20705, USA;
    Ling.Du@usda.gov (L.D.); sangchul.lee@usda.gov (S.L.)
2   Department of Environmental Sciences and Technology, University of Maryland, College Park, MD 20742, USA
*   Correspondence: xia.li@usda.gov (X.L.); greg.mccarty@usda.gov (G.W.M.); Tel.: +1-(301)-504-7401 (G.W.M.);
    Fax: +1-(301)-504-8931 (G.W.M.)

**Abstract:** Landscape topography is an important driver of landscape distributions of soil properties and processes due to its impacts on gravity-driven overland and intrasoil lateral transport of water and nutrients. Rapid advancements in aerial, space, and geographic technologies have led to large scale availability of digital elevation models (DEMs), which have proven beneficial in a wide range of applications by providing detailed topographic information. In this report, we presented a summary of recent topography-based soil studies and reviewed five main groups of topographic models in geospatial analyses widely used for soil sciences. We then compared performances of two types of topography-based models—topographic principal component regression (TPCR) and TPCR-kriging (TPCR-Kr)—to ordinary kriging (OKr) models in mapping spatial patterns of soil organic carbon (SOC) density and redistribution (SR) rate. The TPCR and OKr models were calibrated at an agricultural field site that has been intensively sampled, and the TPCR and TPCR-Kr models were evaluated at another field of interest with two sampling transects. High-resolution topographic variables generated from light detection and ranging (LiDAR)-derived DEMs were used as inputs for the TPCR model building. Both TPCR and OKr models provided satisfactory results on SOC density and SR rate estimations during model calibration. The TPCR models successfully extrapolated soil parameters outside of the area in which the model was developed but tended to underestimate the range of observations. The TPCR-Kr models increased the accuracies of estimations due to the inclusion of residual kriging calculated from observations of transects for local correction. The results suggest that even with low sample intensives, the TPCR-Kr models can reduce estimation variances and provide higher accuracy than the TPCR models. The case study demonstrated the feasibility of using a combination of linear regression and spatial correlation analysis to localize a topographic model and to improve the accuracy of soil property predictions in different regions.

**Keywords:** landscape topography; LiDAR-derived DEM; soil organic carbon; soil redistribution; ordinary kriging; topographic principal component regression kriging

## 1. Introduction

A study of landscape topography is an assessment of the current terrain features and a representation of the landforms. Because topography reflects elevation changes within detailed landform features over a region, it can significantly impact the geomorphological, hydrological, and biological processes on the earth [1]. The spatial variability of topographic features (e.g., relief, slope, and curvatures) controls gravity-driven overland and intrasoil lateral transport of water and nutrients, and impacts soil hydrological regimes, climate, and vegetation types [2].

Topography has been widely used in soil science, with topographic information being derived from multiple sources. Before the 1990s, the main source was geographic maps [1]. Using geomorphometric techniques, the topographic metrics, such as slope gradient and curvatures, were produced manually and applied to investigate spatial variability in soil properties and to produce soil maps [3–5]. With the development of computer and geophysical technologies, more and more scientists have used digital elevation models (DEMs) derived from photogrammetry to calculate topographic metrics. A series of topographic metrics were developed due to the improvement of mathematical theory and physical understanding of topographic surface features.

The objective of this study is to provide an overview of topographic models for predictions of soil properties and processes. The prerequisites for developing an effective topography-based soil property model are that (1) the impacts of topography on soil properties can be investigated through a small set of samplings over a small scale, and (2) strong statistical correlations exist between topography and soil properties. Therefore, this review begins with an introduction of topography-based soil studies, following investigations of topographic metrics that are important for soil models and five main groups of topographic models. The last section presents a case study to assess the efficiency of high-resolution topography-based models for mapping soil organic carbon (SOC) and redistribution.

## 2. Topographic Metrics for Soil Studies

Various studies have demonstrated the utility of including topographic metrics in soil models to better simulate spatial patterns of soil properties and processes [6–15]. Topographic metrics quantify characteristics of the topographic features. According to the calculation methods, topographic metrics can be divided into primary and secondary (combined) metrics [16]. Primary metrics are directly calculated from elevation, such as slope, aspect, and curvatures. The metrics are further grouped into local and nonlocal because of the spatial scope [1]. Local metrics describe the surface geometry at a given point, whereas nonlocal metrics consider relative positions of a selected location. Secondary or combined metrics combine primary metrics and usually describe spatial variability in specific processes such as water content distribution and soil erosion potential. Table 1 lists definitions of topographic metrics that were reported to impact soil movement and properties.

**Table 1.** Definitions of selected topographic metrics.

| Category | | Variable | Definition |
|---|---|---|---|
| **Primary metrics** | Local topographic metrics | Altitude, H (m) | Elevation |
| | | Slope gradient, G (radian) | An angular measure of the relation between a tangent plane and a horizontal plane |
| | | Profile curvature, P_Cur ($m^{-1}$) | Slope change rates in the vertical plane |
| | | Plan curvature, Pl_Cur ($m^{-1}$) | Curvature in a horizontal plane |
| | | Catchment area, CA ($m^2$) | Upslope area contributing runoff to a given point on the land surface |
| | Nonlocal topographic metrics | Upslope slope, UpSl (radian) | Mean slope of upslope area |
| | | Downslope index, DI (radian) | Head differences along a flow path |
| | | Flow path length, FPL (m) | Maximum distance of water flow to a location in the catchment |
| | | Flow accumulation, FA ($m^2$) | Land area that contributes surface water to an area in which water accumulates |
| | | Topographic relief, TR (m) | Elevation difference between the highest point in an area and a given point |
| | | Topographic openness, TO (radian) | Angular measure describing the relationship between surface relief and horizontal distance |
| **Secondary metrics** | | Topographic wetness index, TWI | Frequencies and duration of saturated conditions |
| | | Stream power index, SPI | Erosive power of overland flow |
| | | Length–slope factor, LS | Factor that considers slope length and steepness effects on erosion |

## 2.1. Primary Metrics

### 2.1.1. Local Topographic Metrics

Slope gradient (G), profile curvature (P_Cur), and plan curvature (Pl_Cur) can control gravity-driven overland and intrasoil lateral transport of water and nutrients. G suggests the steepness of a location, which can directly influence water infiltration and soil erosion [17,18]. Compared to relatively flat areas, steeper areas tend to have less infiltration and higher erosion possibilities, decreasing soil moisture and transporting fine soil particles with high SOC content from the areas [2,7]. Furthermore, most of the soil-forming processes (e.g., carbonate dynamics, clay illuviation, and so on) are more efficient on more gently-sloping land surfaces [19,20]. P_Cur is parallel to the direction of maximum slope, and therefore affects soil redistribution and SOC distribution patterns through influencing flow acceleration and deceleration [21–23]. Pl_Cur is perpendicular to the direction of maximum slope, which determines flow divergence and convergence [1,24,25].

Altitude (H) impacts soil properties by affecting climate and insolation. Changes in H cause variations in climate. Generally, decreased temperature and increased precipitation occurs in areas with elevated altitude. Temperature and precipitation changes affect vegetation composition and productivity, which in turn influence soil properties and water content [26].

### 2.1.2. Nonlocal Topographic Metrics

Multiple nonlocal topographic metrics can significantly impact the gravity-driven processes, including upslope slope (UpSl), downslope index (DI), and topographic relief (TR). UpSl reflects the steepness of the upslope contributing areas, which is positively related to overland flow velocities [16,25,27]. DI is also a slope gradient associated metric, but it considers the water balance between the water from a specific upslope contributing area and a downslope area [28]. Therefore, this metric is highly correlated with groundwater gradients and soil water content [28,29]. High values of TR reflect large differences between the highest and the target locations, and thus high overland flow velocities are usually observed with potentials for large downslope soil transport [7,8,30].

Catchment area (CA), flow path length (FPL), flow accumulation (FA), and topographic openness (TO) affect soil properties through influencing soil hydrological regimes, causing variability in soil C decomposition, denitrification, and nitrification processes. Increased CA enhances the chance for sediment deposition, changing the soil C stocks [31]. Longer FPL decreases overland flow velocity and increases soil infiltration and erosion [32–34]. This metric is widely used in soil erosion models because it reflects soil loss under flow divergence and convergence conditions [35,36]. FA impacts water conditions in the soil, which is positively related to flow volume and soil water content [37,38]. TO exhibits convex (high positive TO and low negative TO values) and concave (low positive TO and high negative TO values) landforms [39]. Therefore, soil water contents are likely high in locations with low positive TO values, providing suitable anaerobic environments for denitrification but impeding aerobic C decomposition [7,8].

### 2.2. Secondary Metrics

Commonly used secondary topographic metrics generally address aspects of the physics of water movement on landscapes. Topographic wetness index (TWI) is calculated as $TWI = \ln\left(\frac{CA}{G}\right)$. This index is widely used to reflect the spatial distribution of wetness conditions [7,40–42]. Locations with high values of TWI have high possibilities to be wet locations. TWI has proved to be an effective index for understanding spatial patterns of soil hydrological and geochemical properties and is significantly correlated with soil C and N content [7,8].

Stream power index (SPI) considers specific contributing area ($CA_s$), Pl_Cur ($Pl_{Cur}$), and G with the equation as $SPI = CA_s(Pl_{Cur})\tan(G)$. The metric is useful for investigating potential erosive powers of water flow [43]. The increased G and CA lead to increased water flow velocity and water amount, and consequently enhancing water erosive power [44].

Length-slope factor (LS) is a combination of slope length and slope gradient. There are three major methods for LS calculation, including models developed by Moore and Nieber [45], Desmet and Gover [46], and Wischmeier and Smith [47]. Increased slope length usually increases the soil loss per unit area because of a greater runoff accumulation on a longer slope length, whereas slope steepness increases also stimulate soil loss.

Large collinearities often exist between these topographic metrics (Table 1) for a given landscape, with two main causes for the correlations—one being these metrics quantify the properties of a self-organized landscape whose properties would be expected to be correlated, and another being that various metrics are derived from mathematical equations containing common elements that induce correlations between the resulting metrics. Principal component analysis (PCA) is a common approach to generate sets of orthogonal factors from correlated metrics and thereby reduce the dimension of parameters. These PCA factors, in turn, can be used as a set of orthogonal parameters in prediction models. This is the approach that Li et al. [6–9] used in developing more robust topographic models using information contained within 15 topographic metrics with reduced dimensionality. Interestingly, the resulting PCA factors were combinations of local, nonlocal, and secondary metrics reflecting the connectedness of landscape network processes and information flow.

## 3. Soil-Landscape Models for Soil Property Mapping

Along with the development in computer, aerial, space, and geographic techniques, increasing attention has been paid to soil-landscape modeling to predict spatial patterns of soil morphological, chemical, and physical properties. Recently, detailed large-scale topographic information can be derived due to the increased availability of high-resolution DEMs, providing a possibility for regional-scale soil property predictions based on topography-based models. There are five main groups of models with strong application in landscape modeling.

### 3.1. Geostatistical Models

Geostatistical models deal mainly with spatial data and explain spatial autocorrelation using interpolator. Kriging is a representative geostatistical model with a form of weighted averages generating an estimate from a scattered set of measured values. One limitation of kriging is that it is an interpolation technique, and thus predictions are only valid for regions that have multiple measured values. Estimation in a finite domain can provide too much weight to points, leading to biased estimations [48,49]. In 1980, Burgess and Webster first introduced ordinary kriging (OKr) to map soil textures [50]. Since then, a large body of literature has formed that is based on application of OKr to interpolate soil properties, such as fertility [9], salinity [51], soil water content [52], and infiltration rates [49]. However, OKr fails to consider the knowledge of soil materials and landscape. The efficiency of the prediction usually depends on large samplings at the field scale [49].

To overcome the above OKr limitations, several methods, such as regression kriging (RKr), cokriging (CKr), and kriging with external drift (KED), have been developed to incorporate ancillary data into the OKr model. Combining OKr with multiple linear regression (MLR), RKr spatially interpolates the residuals from a MLR model using kriging and adds the interpolation to the prediction to improve the performance of the MLR model. In soil science applications, various studies have used this method for analyzing spatial patterns of soil horizon thickness [13,14], soil structures [13,14], soil water availability [53], cation exchange capacity [15,54], soil C content [55,56], and hydraulic properties [57,58]. The CKr model takes advantage of correlations between the investigated variable and other easily estimated variables. KED uses external ancillary variables to represent the trend of soil properties. CKr and KED models have also been employed in investigating soil physical and chemical characteristics [12–15]. Several studies have reported that performances of topography-based CKr and KED models are better than OKr models in soil property estimation in areas that are strongly impacted by landscape [49,59].

### 3.2. Logic Models

Fuzzy Logic (FL) is a widely used logical method with soil mapping applications. It is an extension of Boolean logic to express the degree of similarity to a classification type using membership values ranging from 0 (non-membership value) to 1 (membership value) [60]. Because soils are continuums in both geographic and attribute spaces, allowing partial truth of independent variables is especially useful and provides sufficient information about soil properties when compared with the traditional setting of 0 or 1 [61,62]. Topography-based FL has been applied in soil mapping to improve soil taxonomic classification [63–67]. Several studies also used the model to study spatial patterns of soil horizonation [64,68,69], predict soil texture [68,70], and classify soil vulnerability [71].

### 3.3. Decision Tree Analysis

As a divisive supervised classification, decision tree analysis (DTA) successively partitions a dataset into increasingly homogenous subsets. Rules applied to split the data area can be either categorical, such as geographical unit number and soil unit, or continuous, such as elevation and slope [72,73]. A useful rule can decrease impurity of the dataset. Therefore, by developing a set of rules from training data, the DTA can be applied to regions with the same inputs to predict the target variable. This method is useful for capturing nonadditive and nonlinear relationships, and is easier to interpret than standard statistical approaches because the output is based on a set of nominal and/or continuous rules [74].

In accordance with different splitting methods, DTA can be divided into two classes: (1) homogeneous decision tree and (2) hybrid decision tree. The homogeneous decision tree uses a single algorithm in each partition [75]. One representative homogeneous decision tree is the classification and regression trees (CART) algorithm. A topography-based CART can be applied to derive efficient categorical information, such as soil taxonomic classes and soil drainage classification [76–79].

Researchers have also successfully used topography-based CART for quantitatively predicting soil properties such as soil cation exchange and water retention [15,80]. The hybrid decision tree may use different splitting algorithms at different points. Friedl and Brodley [75] found that this method demonstrated the highest accuracy in classification when comparing different DTA methods.

### 3.4. Standard Statistical Methods

Statistical methods, including multiple linear regression (MLR) and discriminant analysis (DA), are widely used to quantify impacts of the landscape on soil properties and to generate soil maps. MLR uses two or more independent variables to simulate a target soil property or process through fitting to a linear equation. The topography-based MLR models do demonstrate, in a quantitative manner, the fact that terrain analysis can be applied to predict spatial patterns of soil physical and chemical properties over large spatial scales [6–12]. MLR has also combined with other methods to improve the efficiency of prediction. For example, Li et al. [7] developed topography-based models based on a principal component regression (PCR) combining principal component analysis and MLR to predict soil redistribution processes and SOC content. Results suggested that the PCR outperformed regular MLR with a more robust prediction over different spatial scales.

DA is a type of supervised classification using categorical criteria to assign an independent variable to the most likely group. The main idea of DA is to develop a set of decision rules on the basis of the measured data using a certain category variable of interest and several auxiliary variables. With establishment of rules, the variable of interest in areas where the auxiliary variables are available can be predicted. This method has been proposed to generate soil texture maps using topographic metrics and other ancillary variables [81,82]. Several studies also demonstrated the feasibility of using the DA to predict soil drainage classes due to its high correlations with topography and soil electrical conductivity [83,84].

### 3.5. Advanced Statistical Methods

Machine learning (ML) is a rapidly developing approach to data analysis that is based on ideas that computers can learn from data, identify trends, and make decisions with limited human intervention. Successful ML methods have several common advantages. ML methods only need a limited number of user-defined parameters. They are able to deal with nonlinear relationships, predict quantitative and category variables, reduce overfitting, and remain robust regardless of outliers [85]. Due to advances in computing power and data availability, increasingly researchers have applied ML in soil mapping applications. Three widely used ML methods on soil property predictions are artificial neural networks (ANNs), deep learning (DL), and random forest (RF) (Figure 1).

ANNs consist of one input layer, one output layer, and one layer of hidden interconnected units (neurons) connecting input and output layers (Figure 1a). With this method, the outputs are related to the input variables, developing linked algorithms of the ANNs model. All links between the input and hidden layers compose the input weight matrix, and links between hidden and output layers are the output weight matrix. The weights can be adjusted iteratively on the basis of the training dataset. Including topographic information and other environmental variables, this method has been successfully applied to identify categorical characters, such as soil taxonomic classes and drainage classification [79,86–89], and to predict quantitative variables including soil chemical and hydrological properties [90–92].

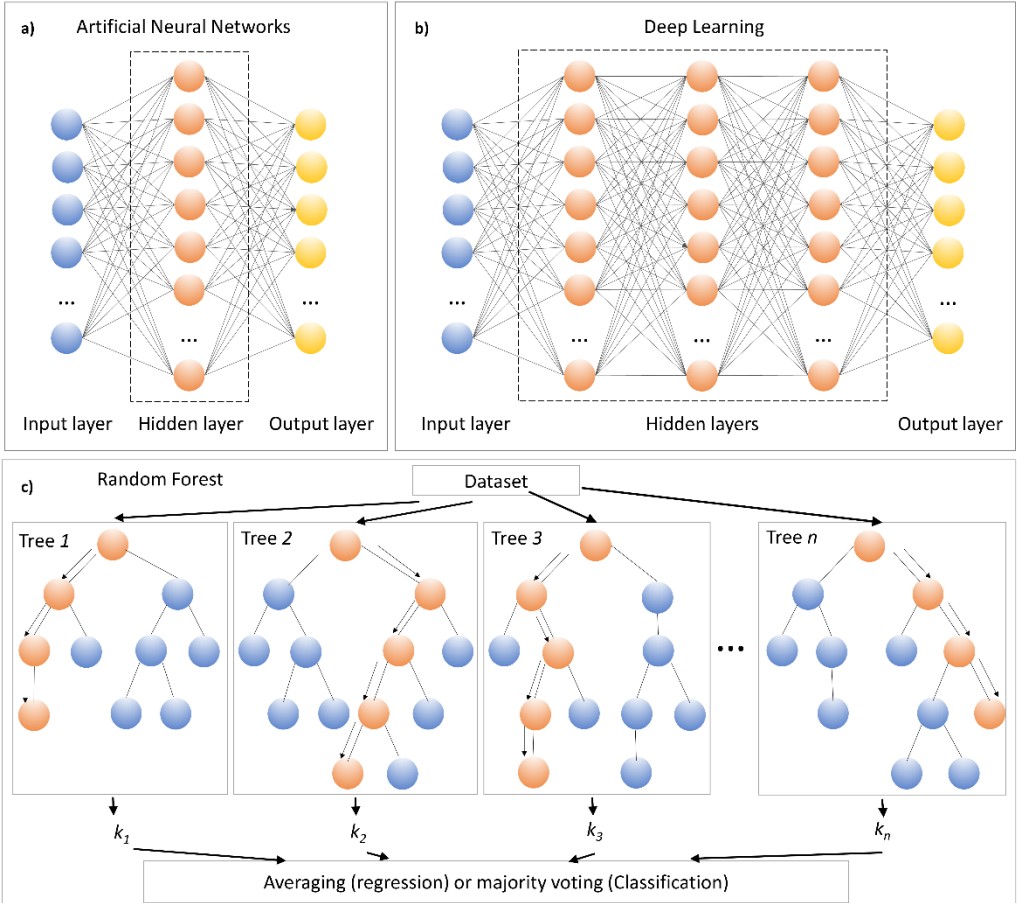

**Figure 1.** Different machine learning architectures: (**a**) artificial neural networks, (**b**) deep learning, and (**c**) random forest.

Advances in computing ability, such as innovative graphics processing units (GPUs), have enabled the use of deep neural networks. DL is considered as an advanced ANN, which includes multi-hidden layers instead of the single hidden layer structure in ANNs (Figure 1b). Learning through multi-layer nonlinear transformations, DL can be used to define edges within images and perform automatic feature extraction [93]. Two major architectures of DL are convolutional neural networks (CNNs) and recurrent neural networks (RNNs). CNNs are based on a layer of convolving window moving along a data array to detect features [94]. Padarian et al. [94] have applied topography-based CNNs to predict SOC at multiple depths and found that CNNs had a lower error than the predictions by the conventional Cubist model. Unlike CNNs, RNNs provide numbers of feedback loops, which allow inputs to be sent to any direction from and to all the layers [95]. As a result, this model has potential advantage for tasks involving sequential information. One study has demonstrated use of RNNs to predict collapse potential of soil (ratio of change in soil height after loading to its initial height) and obtained high accuracy [96].

RF is an ensemble of classification and regression trees. The output of RF can be category estimated by majority voting of the trees or quantitative calculated through the average of the trees (Figure 1c). In the model, each tree randomly selected a subset of features with a random set of training data to increase the diversity of the forest and decrease the correlation of individual trees. Several studies have demonstrated the superiority of RF relative to traditional mathematic methods in soil property predictions due to its high efficiency and low errors [97–100]. Using topographic information as covariates, RF has been successfully applied to predict spatial patterns of soil organic matters and soil texture [99–103] and update soil survey and soil class maps [104–106].

## 4. Case Study

This study used Walnut Creek Watershed (WCW) (41°55′–2°00′ N; 93°32′–93°45′ W), Iowa, as a pilot region to investigate efficiencies of topographic models on predicting soil properties and processes. The studied watershed is in a humid continental climatic zone with a mean annual temperature of 8 °C and mean annual precipitation of 818 mm. The topography of this watershed is relatively flat, with a mean slope of 1.78°. The soils are classified as poor-drained Nicollet (mesic Aquic Hapludolls) soils in the lowlands and well-drained Clarion (mesic Typic Hapludolls) in the uplands [107]. Agriculture is the dominant land-use type. More than 86% of the WCW is croplands. Primary tillage practices are chisel plowing and disking.

Three types of models, including two topography-based models—topographic principal component regression (TPCR) and TPCR kriging (TPCR-Kr) and one geostatistical model of ordinary kriging (OKr), were selected to simulate SOC density and soil redistribution (SR) rate patterns. On the basis of previous reports, we hypothesized that (1) OKr models provide the highest model fit during calibrations; (2) TPCR models provide reasonable regional estimations and can capture spatial patterns of soil parameters outside the area in which the model was developed; and (3) TPCR-Kr models localize TPCR model prediction, which reduces estimation variances caused by model extrapolation.

### 4.1. Methods

#### 4.1.1. Sampling

To test the hypotheses, we collected soil samples from two agricultural field sites in WCW (Figure 2). One (field site 1) is an intensively sampled site in the middle west of the WCW (Figure 2b). A 25 × 25 m grid was created, and 230 soil samples were collected at grid nodes for SOC density and SR rate estimation. Another site (field site 2) is about 4 km east from the field site 1 (Figure 2c). The site includes two 300 m transects with observational SOC density and SR rate. At each location, three samples were collected from 0 to 30 cm soil layer within a 1 × 1 m quadrat using a push probe (3.2 cm diameter). The fields contain both Typic and Aquic soils. Nicollet soils are poorly drained and are located in lower areas and depressions, and Clarion soils are well drained and located in hilltops [107]. Detailed data collection and laboratory analysis can be found in Li et al. [7,25] and Ritchie et al. [22]. A summary of the used data is shown in Table 2. Generally, both sites had negative mean SR rates, suggesting that soils were exported from these two sites within the period of 1960 to early 2000.

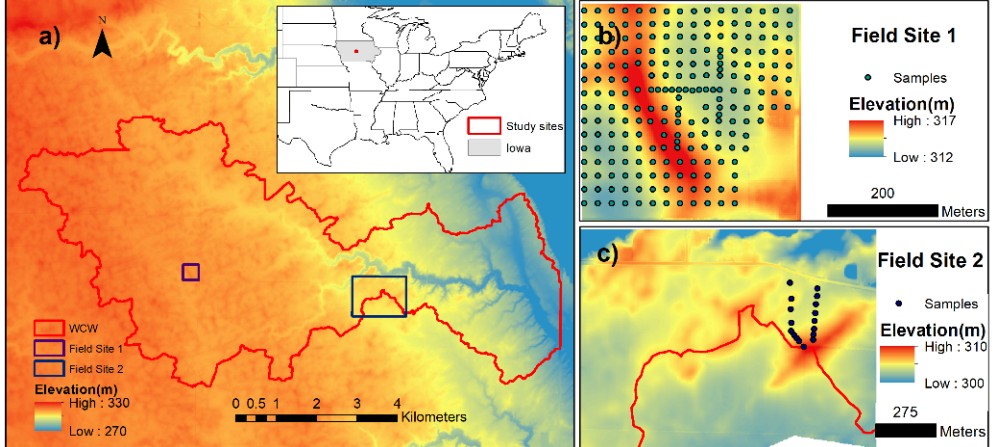

**Figure 2.** Locations of study fields: (**a**) Walnut Creek Watershed (WCW), (**b**) the intensive sampled field site 1, and (**c**) the transect field site 2.

**Table 2.** Mean and standard deviation (SD) of soil organic carbon (SOC) density and soil redistribution rate (SR) at two field sites.

| | SOC (kg m$^{-2}$) | | SR (Mg ha$^{-1}$ year$^{-1}$) | |
|---|---|---|---|---|
| | **Mean** | **SD** | **Mean** | **SD** |
| Field Site 1 | 9.01 | 3.05 | −5.9 | 21.6 |
| Field Site 2 | 7.92 | 3.20 | −4.35 | 30.8 |

### 4.1.2. Terrain Analysis

Light detection and ranging (LiDAR)-derived DEMs were used to generate topographic metrics of the two sites. The LiDAR data were acquired during the period 2007 to 2010 and are available via the Iowa Geodata website. By use of inverse distance weighted interpolation, raw LiDAR data were converted to DEMs with 3 m spatial resolution.

Fourteen metrics that were listed in Section 2 were generated on the basis of the 3 m DEM. The metrics include altitude (A), slope gradient (G), profile curvature (P_Cur), plan curvature (Pl_Cur), catchment area (CA), upslope slope (UpSl), downslope index (DI), flow path length (FPL), flow accumulation (FA), topographic relief component 1 (TRPC1), topographic relief component 2 (TRPC2), positive topographic openness (PTO), topographic wetness index (TWI), and stream power index (SPI). The length—slope factor introduced in Section 2 was excluded due to its high correlation with slope ($r = 0.97$) [7]. We used modules in SAGA to derive slope, P_Cur, Pl_Cur, CA, FA, PTO, DI, FPL, TWI, SPI, and UpSl [6]. Topographic relief was generated on the basis of a maximum elevation map within a specific area and a filtered 3 m DEM [40]. To reduce possible errors caused by an arbitrary selection of a radius for a maximum elevation map, a series of radiuses were selected including 7.5 (relief$_{7.5m}$), 15 (relief$_{15m}$), 30 (relief$_{30m}$), 45 (relief$_{45m}$), 60 (relief$_{60m}$), 75 (relief$_{75m}$), and 90 m (relief$_{90m}$). Principal component analysis (PCA) was applied to convert the relief maps into two independent relief components (TRPC1 and TRPC2).

### 4.1.3. Statistical Analysis and Model Calibration

We used Spearman's rank analysis to analyze topographic impacts on spatial patterns of SOC density and SR rate. Then, we tested performances of OKr, TPCR, and TPCR-Kr models in predicting the above two variables. OKr predicts unsampled locations by weighted averaging of nearby sampled data, and the weights were derived on the basis of semivariogram analysis of the sampled data. For the TPCR models, we first used PCR to analyze topographic metrics of all croplands within the watershed. Components with loadings that explained more than 90% of the variance of all metrics were used to construct topographic principal components (TPCs) used for TPCR models. TPCR-Kr models combine the regression of the dependent variables with the kriging of the regression residuals. Therefore, two steps were included for the TPCR-Kr model implementation: (1) analyzing residuals of TPCR using semivariogram and OKr, and (2) summing the regression prediction and kriging prediction of the residual.

Specifically, OKr and stepwise TPCR models were developed on the basis of SOC and SR observations at the field site 1. The SOC density was log-transformed to meet the residual normality assumption in linear regression. We used the Akaike information criterion (AIC) to select dependent variables in stepwise models. We evaluated the model performances by comparing predictions with observations in the transects. Residuals of the predictions over the transects were calculated and used for developing TPCR-Kr models over field site 2.

Model efficiency was assessed on the basis of three criteria: the coefficient of determination ($R^2$), Nash—Sutcliffe efficiency (NSE), and the ratio of the root mean square error to the standard deviation of measured data (RSR). Generally, the higher the $R^2$ and NSE and the lower the RSR values are, the better the model performs. If the NSE value is larger than 0.5 and the RSR is less than 0.7, the model is considered as satisfactory.

### 4.2. Results and Discussion

4.2.1. Topographic Impacts on SOC and Soil Redistribution

Our results suggested that landscape topography significantly impacted soil properties and soil processes even in relatively flat terrain. TWI was the most influential topographic metric and positively impacted SOC density (Table 3). It also showed a high positive correlation with SR rates (Table 3). Soil water conditions that TWI reflects can be a reason for these high correlations [40]. Generally, areas with low TWI values tend to be drier than areas with high values. Therefore, the aerobic environments in low water content areas would support rapid aerobic decomposition of soil C, leading to low SOC storage [7,9,108,109]. Meanwhile, a location with high TWI usually suggests that the site has a large catchment area and low slope gradient, which would promote sedimentation of fine particles with high proportions of SOC content [110,111]. This further explained the high positive correlations between TWI and SOC density, as well as TWI and SR rates in this study.

**Table 3.** Topographic impacts on soil organic carbon (SOC) and soil redistribution (SR) rate.

|  | A | G | P_Cur | Pl_Cur | CA | UpSl | DI | FPL | FA | TRPC1 | TRPC2 | PTO | TWI | SPI |
|---|---|---|---|---|---|---|---|---|---|---|---|---|---|---|
| **SOC** | −0.441 *** | **−0.669** *** | −0.212 *** | −0.336 *** | **0.537** *** | −0.289 *** | 0.432 *** | 0.419 *** | −0.225 *** | **0.665** *** | −0.150 * | −0.583 *** | **0.722** *** | - |
| **SR** | −0.441 *** | **−0.591** *** | −0.225 *** | −0.248 *** | **0.513** *** | −0.170 ** | 0.404 *** | 0.437 *** | −0.196 ** | **0.633** *** | - | −0.488 *** | **0.605** *** | 0.128 * |

Note: *** $p < 0.0001$, ** $p < 0.005$, * $p < 0.05$. A is altitude; G is slope gradient; P_Cur and Pl_Cur are profile curvature and plan curvature, respectively; CA is catchment area; UpSl is upslope slope; DI is downslope index; FPL is flow path length; FA is flow accumulation; TRPC1 and TRPC2 are topographic relief principal components 1 and 2, respectively; PTO is positive topographic openness; TWI is topographic wetness index; and SPI is stream power index. The value in bold is correlation coefficient >0.5, and the value in red and bold indicates the highest correlation coefficient for each soil property.

Topographic relief (TR) presented the highest positive correlation with SR rate and a high positive correlation with SOC density. TRPC1 was mainly related to the large-scale relief maps (relief$_{30m}$, relief$_{45m}$, relief$_{60m}$), which exhibited landscape fluctuation over a large area, whereas TRPC2 was dominated by the small-scale relief maps (relief$_{7.5m}$), showing location variation at a small location. The substantial impacts of TRPC1 on SR rates and SOC density may have been due to its influences on flow velocity. Areas with high TRPC1 values have large elevation differences from the most top points, which accelerate flow velocity, causing more transport of soil from low relief (small elevation difference) to the high relief (large elevation difference) areas [112,113]. Therefore, SR rates and SOC density values were high in high TR locations.

Several other topographic metrics, such as G and CA, were also highly related to SOC density and SR rate ($r > 0.5$). G and CA were related to flow velocity and flow accumulation, respectively. This finding is similar to the study of Fox and Papanicolaou [114], which found that overland flow regulated the incoming water and soil from uplands in a low-relief agricultural watershed.

4.2.2. Topography-Based Model Evaluations

Both TPCR and OKr models obtained satisfactory results in predicting SOC density and SR rate at the field site 1, with high NSE (>0.5) and low RSR (<0.7) values (Figure 3). All the generated maps showed high SOC density and SR rates in depressions and low values in sloping and ridge areas (Figure 4). Specifically, OKr models provided the best simulations with $R^2$ up to 0.791 for SOC density and 0.947 for SR rate. Spatial patterns of OKr SOC and OKr SR were highly consistent with bare soil image and elevation map over the field scale (Figure 4a,b,d,e).

The better performances of OKr over TPCR-Kr supported our first hypothesis. OKr is regarded as a method of optimal, unbiased, and minimal variance estimation, providing analyses of spatial correlation in the data [115]. Similar results were also reported by Eldeiry and Garcia [51], which found that OKr outperformed regression kriging and cokriging in soil salinity estimation. However, OKr also has some limitations. It has been criticized for failing to consider soil processes and environmental

variables. This method often needs a large number of samples that are close to each other to improve the accuracy of prediction. Decreases in sample numbers can rapidly increase the estimation variances [49]. Furthermore, the OKr model is a spatial interpolator. Applying to regions outside the area that model developed would cause high prediction errors [116].

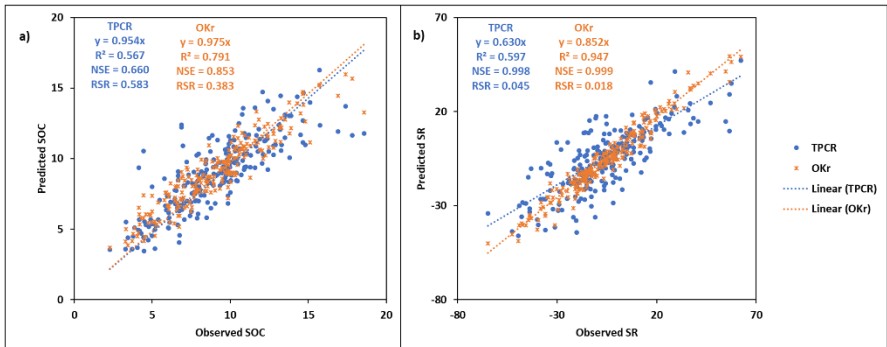

**Figure 3.** Comparison of topographic principal component regression (TPCR, blue data points) and ordinary kriging (OKr, orange data points)-derived (**a**) soil organic carbon (SOC) density (kg m$^{-2}$) and (**b**) soil redistribution (SR) rate (Mg ha$^{-1}$ year$^{-1}$) estimations for observations at field site 1.

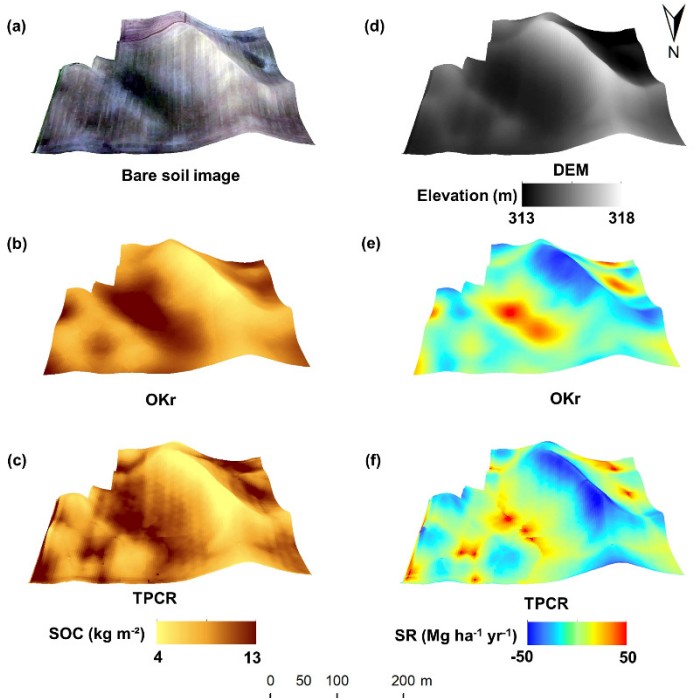

**Figure 4.** Maps of landscape characteristics and model-derived soil property maps at field site 1: (**a**) bare soil image; (**b**) ordinary kriging (OKr)-derived SOC density; (**c**) TPCR-derived SOC density; (**d**) elevation; (**e**) ordinary kriging (OKr)-derived soil redistribution (SR) rate; and (**f**) TPCR-derived SR rate.

The TPCR models have inherent advantage over OKr models in that they are not limited to data interpolation because they are based on topographic metrics to a much larger region where high resolution topographic data are available (Figures 5 and 6). The predicted values explained 67% and 76% of the variability in transect SOC density and SR rate, respectively (Figure 5). High consistencies between bare soil image and SOC and SR maps demonstrated the efficiencies of using the regional TPCR models developed at field site 1 to detect the spatial variability of soil parameters at field site 2 (Figure 6). However, the two TPCR models tended to underestimate the observations over

transects. One possible reason for the deviations that existed after extrapolation may be related to field management. Each production field has unique management history that can affect storage of SOC and movement of sediments over the landscape [117,118].

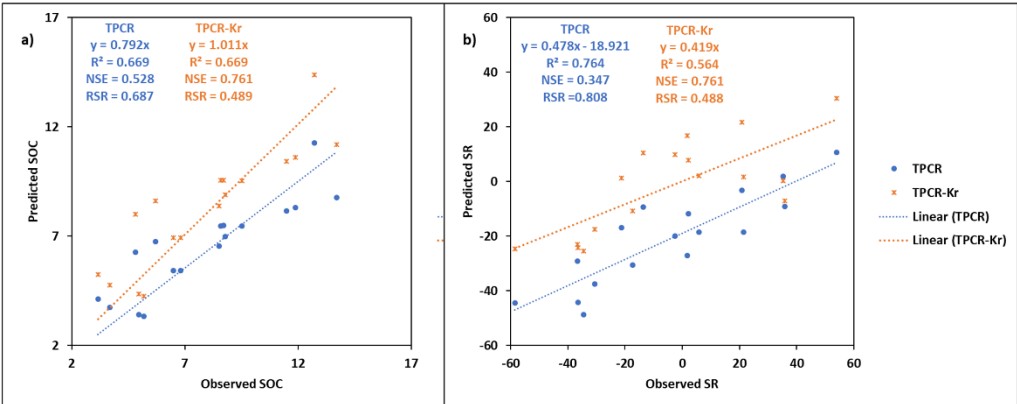

**Figure 5.** Comparison of topographic principal component regression (TPCR, blue data points) and TPCR-kriging (TPCR-Kr, orange data points)-derived (**a**) soil organic carbon (SOC) density (kg m$^{-2}$) and (**b**) soil redistribution (SR) rate (Mg ha$^{-1}$ year$^{-1}$) predictions for observations at field site 2.

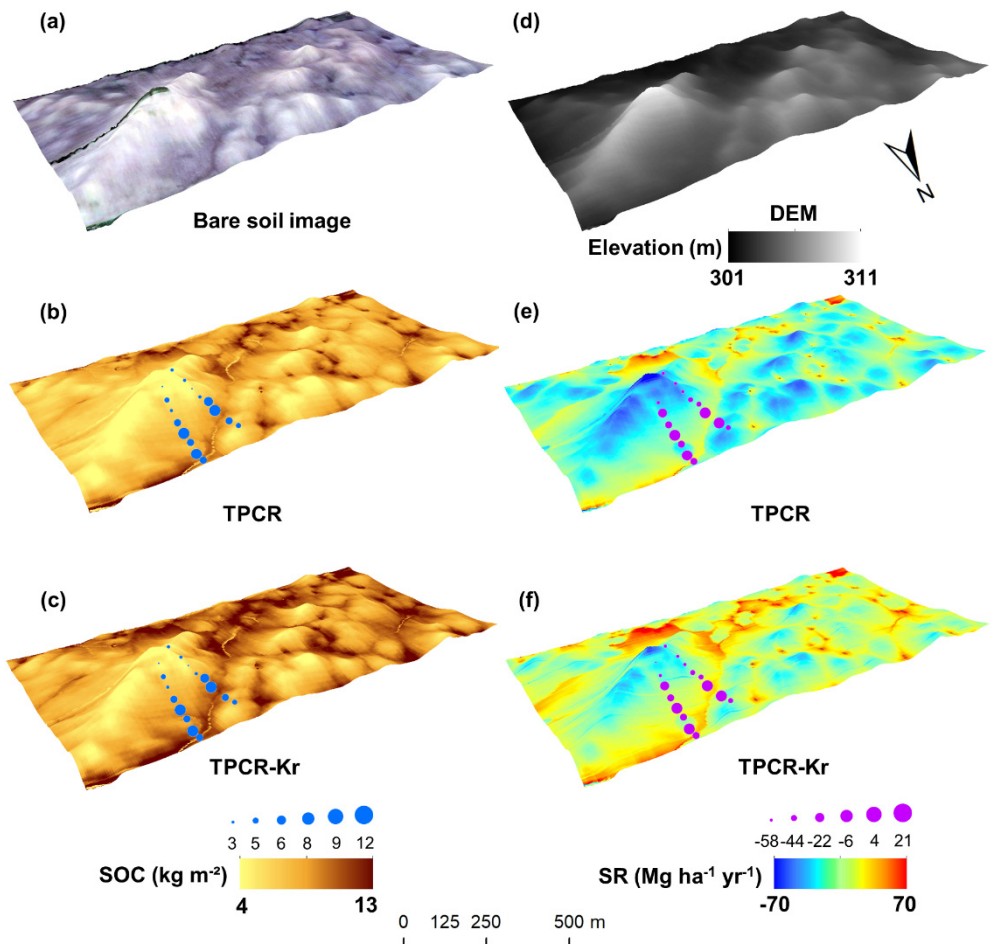

**Figure 6.** Maps of landscape characteristics and model-derived soil property maps at field site 2: (**a**) bare soil image; (**b**) topographic principal component regression (TPCR)-derived SOC density; (**c**) TPCR-kriging (TPCR-Kr)-derived SOC density; (**d**) elevation; (**e**) TPCR-derived soil redistribution (SR) rate; and (**f**) TPCR-Kr-derived SR rate.

TPCR-Kr models reduced the estimation variances and were better fitted for the soil property and process predictions in transects, which substantiated our third hypothesis. Specifically, the TPCR-Kr SOC model provided the same range as observations with a slope close to 1. The NSE values increased from 0.528 to 0.761 with use of TPCR and TPCR-Kr SOC models, respectively, and RSR values decreased from 0.687 to 0.489 with use of TPCR and TPCR-Kr SOC models, respectively. Use of the TPCR-Kr SR also improved the accuracy relative to the TPCR SR model, as presented by the increased NSR and decreased RSR values.

The improvement of the TPCR-Kr models suggested a possible use of local calibrations for correcting regional TPCR models. A cropland field within a region has a unique management history, which can increase deviations in measured soil properties from a soil map generated by a regional TPCR model [117,118]. The spatial patterns of the soil properties under topographic influences as produced by TPCR may be less variable than the estimated ranges for the observed parameters. As a result, TPCR can produce accurate patterns of soil property distribution but the means and ranges of the properties may be inaccurate. A relatively small number of samples could be collected from the field of interest to adjust the topographic model using TPCR-Kr to better reflect the range in measured values of the soil parameter. With this approach, map accuracy would improve.

In summary, the advantage of topographic models is the ease at which necessary data can be acquired over large geographic areas by means of LiDAR mapping. Visual inspection of the bare soil image and the regional TPCR map for SOC demonstrated remarkable high fidelity for a regional model predicting spatial patterns of SOC on a distant cropland field with fine-scale detail. Application of TPCR-Kr with inclusion of a relatively small number of local samples can localize the calibration for an area of interest while retaining the high-fidelity spatial patterns.

Principal component analysis (PCA) of the geographic region can be performed independent of physical sampling for soil characteristics. Sampling strategies can then be developed such that the full range of principal components (PCs) are covered within the sample set. PCA also affords the ability to test that a location is within the domain of the topographic model by use of repeated subsampling of a landscape to generate a population of PC sets for statistical testing of differences from PCs of the modeling domain. Li et al. [6] demonstrated that when topographic models are extrapolated to a watershed scale, the TPCR models are superior to ordinary least square regression models, likely because of reduced overfitting during calibration. The PCs used in the TPCR models were developed at the watershed scale, providing better assurance that model parameters reflected the topography of the larger setting.

## 5. Conclusions

This report reviewed the effects of landscape topography on soil properties and processes and introduced five main groups of topography-based models. We then highlighted three models—OKr, TPCR, and TPCR-Kr—by comparing the performances in SOC density and SR rate interpolations and extrapolations in a case study. OKr provides analyses of spatial relations and TPCR considers topographic information. Although the OKr models provided the best fits for SOC and SR calibrations, the high accuracies require a large set of field sample data, which is often not available for many regions. In contrast, the TPCR models utilized data only from remotely sensed data, providing cost-effective methods to investigate soil spatial patterns. However, variations in climate, environment, and human management may influence soil properties and processes, increasing estimation variances when applying the regional TPCR model to different fields of interest [119–121]. TPCR-Kr improved the model accuracy by including sample correction. The application of TPCR-Kr allowed local calibration of the TPCR model to better reflect the mean and range of measured soil parameters. With a very limited number of new samples, a regional topographic model can be adjusted to the local condition, and spatial patterns of soil properties can be mapped with high fidelity.

**Author Contributions:** Conceptualization, X.L. and G.W.M.; methodology, X.L. and L.D.; writing—original draft preparation, X.L.; writing—review and editing, G.W.M., S.L., and L.D; supervision, G.W.M.; project administration, G.W.M.; funding acquisition, G.W.M. All authors have read and agreed to the published version of the manuscript.

**Funding:** This research was funded by the United State Department of Agriculture Natural Resources Conservation Service in association with the Wetland Component of the National Conservation Effects Assessment Project (NRCS 67-3A75-13-177).

**Acknowledgments:** This research was supported by the United State Department of Agriculture Natural Resources Conservation Service in association with the Wetland Component of the National Conservation Effects Assessment Project (NRCS 67-3A75-13-177).

**Conflicts of Interest:** The authors declare no conflict of interest.

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
