# Peer review of "Use of Topographic Models for Mapping Soil Properties and Processes"

_soilsystems, doi:10.3390/soilsystems4020032_

Round 1
Reviewer 1 Report
The ms soilsystems-779167, "Use of topographic models for mapping soil 2 properties and processes" by Xia Li1 et al. presents present a summary of recent topography-based soil studies and review five main groups of topographic models used in geospatial analyses widely used for soil sciences. In detail the Authors compared performances of two types of topography-based models - topographic principal component regression, and kriging to ordinary kriging models in mapping spatial patterns of soil organic carbon density and redistribution rate. Therefore in my oppinion it fits the goals of Soil Systems journal, it may be of interest to a vast community of users and deserves being published in this Journal.
Author Response
We appreciate the reviewer for his/her encouragement. We made a few modifications and believed the revised version further improved the manuscript. Thanks again for your review.
Reviewer 2 Report
GENERAL COMMENTS
The paper compares different models in mapping soil organic carbon content and soil redistribution. The topic of the paper is not novel, but of practical and scientific interest and it could contribute useful data to the knowledge base. The paper is well written and organized. English is fine and minor spell check is required. Figures are clear and adequate to illustrate the key-concepts and the main results. These latter are presented and discussed in the same section, but data are quite clearly separated from interpretations. Study area should be described in more detail (see Specific Comments below). The Introduction section is well written and organized and provides an exhaustive description of the state-of the-art, even if in a synthetic way, given the huge amount of existing papers on topography-based soil studies. This section reviews also models not used by the Authors in the presented case study. Some other suggestions are listed in the Specific comments below. In my opinion, manuscript is suitable for publication on Soil System with minor revisions, pending other Reviewers’ and editorial comments.
SPECIFIC COMMENTS
Page 1, Line 36 - Better "landforms"
Page 3, Table 2 - I suggest refining the definition of SL factor in the table, as the provided one does not take into account the slope gradient, which is central in defining and calculating LS factor.
Page 3, Lines 73-75 - I would also quote the fact that most of the soil-forming processes (e.g., carbonate dynamics, clay illuviation and so on) are more efficient on flat or gently-sloping land surfaces.
Page 3, Line 94 - Better "soil properties" (here and throughout the text).
Page 3, Line 99 - Do you mean "positively"?
Page 4, Line 147 - Better "huge amount of publications".
Page 4, Line 152 - "...a MLR model"
Page 5, Line 205 - Better "soil texture maps"
Page 6, Line 222 - Do you mean "connecting input…"?
Page 7, Lines 251-257. Some more information would be helpful to better characterize the study area, e.g. rainfall, temperature, slope angle range and main aspect classes.
Page 7, Line 252. What is the watershed extension?
Page 7, Lines 254-256. Is that an original classification of soils? Otherwise, proper references should be quoted.
Page 7, Lines 258-259. Better "...two topography-based models, i.e. principal component regression (TPCR) and TPCR kriging (TPCR-Kr), …"
Page 7, Line 260. Better "...were selected..."
Page 7, Line 261. 1) Better "hypothesise"; 2) On which basis do the Authors hypthesize this? Literature data?
Page 7, Line 269. Which is the soil sampling depth interval?
Page 7, Lines 268-270. Which kind of Hapludoll (Typic or Aquic) is present in the two field sites?
Page 7, Lines 273-274. What do you mean with "were transported"? Please clarify.
Page 8, Line 276 (Figure 2) - WCW should be more accurately delineated on geomorphological basis; otherwise, it is not a watershed, but a part of a watershed.
Page 8, Line 297. The acronyms of Topographic Relief Components were RePC1 and RePC2 at line 288 and TRPC1 and TRPC2 here (line 297). Are you talking about the same components? In this case, to avoid confusion, the same acronym should be used. Please clarify.
Page 8, Line 303. Do you mean "over the whole WCW"?
Page 9, Line 327. Do you mean "...tend to be drier"?
Page 9, Lines 348-349. ...Transport of soil from low relief to high relief areas?
Page 12, Line 393. Specify, in the figure caption, what do the blue and violet dots indicate.
